# Surfactant Protein B Plasma Levels: Reliability as a Biomarker in COPD Patients

**DOI:** 10.3390/biomedicines11010124

**Published:** 2023-01-04

**Authors:** Michela D’Ascanio, Fausta Viccaro, Dario Pizzirusso, Giulio Guerrieri, Alessandra Pagliuca, Simone Guerrini, Marta Innammorato, Claudia De Vitis, Salvatore Raffa, Aldo Pezzuto, Alberto Ricci

**Affiliations:** 1Department of Respiratory Diseases, Azienda Ospedaliera Sant’Andrea, Sapienza University, Via di Grottarossa 1035/1039, 00189 Rome, Italy; 2Department of Clinical and Molecular Medicine, Sapienza University, 00189 Rome, Italy

**Keywords:** surfactant protein B, FEV1, COPD

## Abstract

Background: The diagnosis of COPD is based on both clinical signs and functional tests. Although there are different functional tests used to assess COPD, no reliable biomarkers able to provide information on pathogenesis and severity are available. The aim of the present study is to explore the relationship between surfactant protein B (Sp-B) serum levels and clinical, radiological, and functional pulmonary parameters in COPD patients. Methods: Forty COPD patients and twenty smokers without airflow limitations or respiratory symptoms were enrolled. Each patient was given questionnaires (CAT and mMRC) and 6MWT, spirometry, DLCO, and computer tomography (CT) were performed. All participants underwent a venous blood sample drawing, and quantitative detection of their Sp-B plasma levels was performed by an enzyme-linked immunosorbent assay. The spirometry and Sp-B plasma levels were assessed after 12 months. Results: A statistically significant difference was found in the plasma Sp-B levels between COPD patients compared to the other group (4.72 + 3.2 ng/mL vs. 1.78 + 1.5 ng/mL; *p* < 0.001). The change in FEV1 after 12 months (Delta FEV1) showed a significantly negative correlation with respect to the change in Sp-B levels (Delta SpB) (r = −0.4; *p* < 0.05). This correlation indicates that increasing the plasma dosage of SpB is a foretoken of functional decline. Conclusions: SpB may be considered as a useful marker in COPD assessment and provides prognostic information on lung functional decline. Despite its usefulness, further studies are needed to define its reliability as a biomarker.

## 1. Introduction

Chronic obstructive pulmonary disease (COPD) is a common worldwide disease—it is the third leading cause of death [1]. The diagnosis of COPD is based on clinical signs such as cough, sputum production, dyspnea, previous exposure to inhalation risk factors, and/or frequent respiratory exacerbations. Functional tests are characterized by irreversible airflow limitation (FEV1/FVC < 0.70) post-bronchodilator during a forced expiration test [2,3,4]. Forced expiratory volume in the first second (FEV1) is the most widely used marker of disease severity and progression, but it poorly correlates with both the symptoms and other measures of disease progression. Despite all these validated tools, some aspects of COPD remain unknown. Therefore, it could be useful to have validated biomarkers for early diagnosis and prediction of progression, resulting in consequent therapeutic implications.

Currently, no serum biomarker has been validated, though research is moving in this direction. However, some studies of some blood-based markers have shown promising results [3].

Biomarkers are defined as objectively measurable features indicative of normal biological processes, pathogenic processes, and prognosis, predicting pharmacologic responses to a therapeutic intervention. The development of biomarkers that identify the endotypes most likely to respond to targeted pharmacological treatments is essential for precision medicine [5].

For practical purposes, most published studies have analyzed blood biomarkers because of their easy accessibility and assay reproducibility. In line with these findings, the purpose of our study was to analyze the possible role of plasma surfactant protein B (SP-B) levels in COPD.

Pulmonary surfactant has important functions beyond reducing surface tension to avoid lung collapse during respiration and provides a crucial first line of defense against infection [6]. Pulmonary surfactant is a complex mixture of lipids (90%) and proteins (5–10%) that constitutes the mobile liquid phase covering the large surface area of the alveolar epithelium. Four surfactant proteins (SPs), SP-A, SP-B, SP-C, and SP-D, are intimately associated with surfactant lipids in the lung. Their importance is demonstrated through studies performed in the therapeutic field through the administration of exogenous surfactant [7,8]. The properties of SP-B have made it a useful protein to study within the context of lung diseases such as acute respiratory distress syndrome, neonatal respiratory distress, and lung cancer [9,10]. However, few studies have explored its role in COPD, especially surfactant protein B (SP-B). In fact, most studies on surfactant proteins have been performed on idiopathic pulmonary fibrosis (IPF). SP-A and SP-D are the proteins that, more than any others, have shown greater reliability as a possible biomarker of disease because of their characteristics and reproducibility in determination. Serum SP-A/D detection might be useful for the differential diagnosis and prediction of survival in patients with IPF [11].

SP-D expression has also been studied in COPD patients [5]. It is statistically associated with emphysema progression and mortality in the COPD Gene cohort, but the influence of age, body mass index, sex, and current smoking status is not clearly defined, making SP-D plasma levels difficult to use as a marker for any treatment effects or prognostic outcome [5].

In lung cancer, plasma Pro-SP-B represents an independent predictor of lung cancer and represents an additional marker to the existing ones that analyze the risk of developing disease. The same authors demonstrated that mature SP-B was increased in patients with resectable non-small cell lung cancer (NSCLC) relative to controls [10].

To our knowledge, few studies have been conducted on serum SP-B concentration and COPD, and some of these show conflicting results. The reason could be due to SP-B’s hydrophobicity, which does not make it easily determinable. Its mature form is an 8 kD hydrophobic molecule exclusively synthesized by type II pneumocytes and non-ciliated intra-alveolar bronchiolar cells [12]. SP-B has higher specificity for the lungs, because the lungs exclusively produce this protein [13].

Due to poor solubility in the plasma and broncho alveolar lavage (BAL), some studies have used its precursor, pro-SPB, which is a hydrophilic 42-kD protein that is cleaved at the N and C terminus into its mature form and secreted into the alveolar space [12].

The aim of this study is to evaluate the usefulness of SP-B as a potential biomarker in COPD patients. Therefore, we investigated whether SP-B plasma levels were modified in COPD patients in relationship with heathy controls and whether these changes were related to the clinical, functional, or radiological findings.

## 2. Materials and Methods

### 2.1. Study Design and Cohort

The study was an observational, cross-sectional, monocentric study conducted at Sant’Andrea Hospital, in Rome. Patients enrolled were recruited in 12 months run-in period, from December 2018 to December 2019. This study was approved by the ethics committee of Sant’Andrea Hospital, Rome (5078_2018). Each participant provided written informed consent. This study was conducted according to the Declaration of Helsinki and good clinical practice guidelines.

Forty patients with stable COPD, as defined by GOLD 2022, and twenty smokers without respiratory symptoms and or spirometry abnormalities were studied. Patients were recruited who met the inclusion criteria: (a) ≥45 years old; (b) post-bronchodilator FEV1/FVC < 0.7; (c) smokers or former smokers (at least 10 pack/years); (d) current bronchodilator therapy. The exclusion criteria were (a) history of a recent (less than 4 weeks) respiratory tract infection; (b) other chronic respiratory diseases (i.e., cystic fibrosis, severe bronchiectasis, previous lobectomy, and restrictive lung disease); (c) severe comorbidities (cancer and unstable cardiovascular diseases); (d) low compliance to perform all functional tests.

### 2.2. General Measurements

In all participants, the following parameters were collected during the enrollment visit: anthropometric variables (age, sex, and body mass index (BMI)), smoking habit (current/former), and number of packs per year. In COPD patients, number of previous exacerbations per year, COPD Assessment Test (CAT) score, modified Medical Research Council dyspnea scale (mMRC), and six-minute walking test (6MWT) were also evaluated.

### 2.3. Spirometry, Plethysmography, and Diffusing Capacity for Carbon Monoxide

Patients were advised to regularly assume their inhaler therapy. Lung function tests were performed with automated equipment (Master Screen Body PFT Jaeger, Wurtzburg, Germany) using the current recommendation of ATS/ERS Task Force on standardization of pulmonary functions test [14,15]. At least 3 measurements were taken from each spirometry and lung volume to assure reproducibility. To collect lung volumes FEV1, total lung capacity (TLC), and residual volume (RV), a pneumotachograph and whole-body plethysmography were used. Pulmonary diffusing capacity for carbon monoxide (DLCO) determined by the single breath technique was performed. DLCO expresses the capacity of the lung to exchange gas across the alveolar–capillary interface, determined by its structural and functional properties [16]. Spirometry was performed both at baseline (T0) and after 12 months (T1).

### 2.4. Chest Computed Tomography

All patients underwent baseline chest computed tomography (CT). A qualitative emphysema analysis assessment was scored by using the modified method from Kazerooni et al. [17]. Each of the five lobes of the lung plus the lingula was scored on a scale from 0 to 4 points, defined as follows: 0 = absence of emphysema, 1 = 1–25% emphysema, 2 = 26–50% emphysema, 3 = 51–75% emphysema, and 4 = 76–100% emphysema. The total score was made by the sum of each lobe.

### 2.5. Specimen Handling and Assay

Before starting clinical sessions, each participant underwent a venous blood sample drawing.

Patients underwent further sampling after 12 months. Fresh blood was collected in vacutainer tubes containing citrate 0.129 mol/L as an anticoagulant, centrifuged at 3000 rpm at 4 °C, and divided into aliquots, and plasma was stored at −80 °C before the analysis. The quantitative analysis of SP-B levels was performed by an enzyme-linked immunosorbent assay (SEB622Hu), as previously described [18,19]. The concentration of SP-B in the samples was then determined by comparing the absorbance of the samples to the standard curve and expressed as ng/mL.

### 2.6. Statistical Analysis

Dichotomous variables were presented as proportions and continuous variables as mean + SD. Fisher’s exact test or Student’s *t*-test was used as appropriate. Pearson’s chi-squared test was used for the correlation analysis.

A two tailed *p* < 0.05 was considered to indicate statistical significance. Analysis was performed using SPSS.

## 3. Results

This study cohort included 40 stable COPD adult patients and 20 smokers as subjects. Baseline characteristics for all subjects are reported in Table 1. The mean age of the COPD patients was higher compared to the other group (71 + 8 vs. 65 + 7; *p* < 0.05), and in both groups there was a prevalence of men (75% and 65%, respectively). All subjects were smokers or former smokers.

There was a significant difference in SP-B plasma levels between healthy smokers and COPD patients (1.78 + 1.5 ng/mL and 4.72 + 3.2 ng/mL, respectively; *p* < 0.001) (Figure 1). The mean FEV1 and TLC values (expressed as percentages) of the COPD patients were, respectively, 57.6 + 21, 97.6 + 14, and 78 + 27 for DLCO/VA. No significant correlation was found between SP-B plasma levels and FEV1, TLC, RV, or DLCO/VA.

Fifteen patients in the COPD group had signs of emphysema on a CT scan. We also analyzed the correlation between the levels of emphysema on CT and SP-B, but no statistically significant correlation was found.

The delta FEV1 and delta SP-B obtained from the difference in their respective values were calculated after 12 months and at baseline (T1-T0). Changes in FEV1 after 12 months (T1-T0) showed a statistically significant negative correlation with SP-B plasma levels (r = −0.415; *p* < 0.05) (Figure 2).

A direct correlation was also found between SP-B plasma levels and BMI (r = 0.3; *p* < 0.05). On the contrary, no correlation was found between the SP-B plasma levels and clinical findings, especially with the number of exacerbations and symptoms or with the score obtained by completing the CAT and mMRC questionnaires.

## 4. Discussion

### 4.1. SP-B as a Biomarker

Circulating surfactant protein-B levels are increased in COPD patients in comparison to healthy smokers. Furthermore, plasma SP-B levels are related to COPD clinical status. The novel and unique ability of plasma SP-B to reflect lung damage is supported by its association with the decline of FEV1 and suggests that SP-B may be a useful prognostic biomarker in COPD.

However, some results appeared conflicting. In fact, no significant correlation was found between serum SP-B concentration and the functional, radiological, or clinical data. To our knowledge, this is the first study that explored SP-B levels, COPD lung function, and the treatable traits (symptoms and exacerbations).

According to Leung’s studies, we expected to find a negative correlation between SP plasma levels and FEV1 [20]. Our data appear to be in contrast with this study; a possible explanation is that SP-B, in its mature form, is a less reliable index of pathology, probably due to its poor solubility [12]. Another possible explanation is that in stable COPD patients (particularly those with lower FEV1), anatomical remodeling of lung parenchyma has already occurred. Therefore, it is possible that this condition, where the anatomical damage has already consolidated, is associated with a reduced spillover of proteins from the alveolar cells into the bloodstream. This would also explain the low levels of SP-B observed in the BAL.

In accordance with this hypothesis is the observation that lower BAL SP-D levels were associated with worse lung function in COPD subjects [13,21].

The increase in plasma SP-B levels in COPD agrees with previous investigations, though there are many controversial findings as well. Pro-SFTPB concentrations in BAL fluid were significantly related to lung function, and its expression could be increased with the use of an inhaled budesonide/formoterol combination [21]. The same authors also highlighted how pro-SFTPB in plasma was not related to airflow limitation and was not responsive to short term therapy. This likely occurred because the plasma expression of pro-SFTB is relatively low even in patients with COPD, reducing its discriminative property as a blood biomarker [21].

In COPD patients, both during acute exacerbation and the stability phase, no significant changes were observed in the levels of any SPs between admission and discharge or the stability in the overall population or among the subgroups: smokers and non-smokers [21].

On the contrary, in studies conducted on acute lung damage such as ARDS, acute pulmonary edema, and acute lung failure, high levels of SP-B isoforms are measurable in blood and configure damage at the level of the alveolus capillary barrier, proving to be of great use in clinical practice.

### 4.2. SP-B as Prognostic Factor

Another important result of our study is the prognostic role of SP-B. The correlation between delta Fev1 and delta SP-B indicates that increasing SP-B plasma levels is a foretoken of functional decline.

The mechanisms that determine the increased plasma concentration of SP-B when there is lung damage remain to be elucidated. The main function of SP-B is to accelerate the formation of a surface-active film composed of phospholipids at the air–water interface, by increasing the adsorption rate. SP-B also has anti-inflammatory properties and may be involved in protecting the lung against oxidative stress. With smoking or acute lung injury, lung BAL SP-B protein concentration decreases. Most importantly, to our knowledge, no extra-pulmonary organs produce any appreciable amount of SP-B, making it a highly specific lung biomarker [21].

Surfactant proteins are transcriptionally regulated by thyroid transcription factor 1 (TTF-1)/NK2 homeobox 1 (NKX2-1), which plays a crucial role in lung development. TTF-1/NKX2-1 is markedly increased in regions of the lung parenchyma undergoing regeneration and repair. In addition, increased expression of TTF-1/NKX2-1 in alveolar type II cells has been found to induce dose-dependent alterations in alveolar morphology, epithelial cell hyperplasia, emphysema, and pulmonary inflammation [22,23].

Probably the emphysema and, consequently, the damage of the alveolar–capillary barrier are associated with the release of SP-B from the lung into the bloodstream. However, given the important role that SP-B plays in the regulation of surface tension, it cannot be excluded that smoking-related lung damage may be responsible for altered SP-B conformation that, in turn, facilitates its release and spillover. The reduction in SP-B levels at the alveolar level leads to the worsening of respiratory symptoms for early alveolar collapse, which is related to the loss of its regulatory role in the stabilization of surface tension. It is known that hereditary, in full-term infants, or maturational, in pre-term infants, SP-B deficiency is usually present in respiratory distress syndromes [20].

While apoptosis is thought to be the mechanism by which emphysema develops, an additional role may be played by SP-B, particularly considering its role in modulating inflammatory micro-environment. SP-B deficiency in a transgenic murine model increases pulmonary inflammation, including the increase in the BAL fluid total cell number, macrophage and neutrophil migration into the lung, and levels of interleukin (IL)-6, IL-1β, and macrophage inflammatory proteins. This heightened inflammatory response is, furthermore, attenuated to the restoration of SP-B levels. Similarly, SP-B, unlike SP-A and C, may modulate the oxidative stress response of the lung. Specifically, in vitro models of alveolar macrophages cultured from rat lungs, when incubated with SP-B, showed reduced nitric oxide levels in response to lipopolysaccharide [24]. Both the anti-inflammatory and anti-oxidative properties of SP-B may act to protect the lung from toxic agents and cigarette smoke [24].

SP-B is also considered an accurate indicator of barrier damage [25], as detected in heart failure. Moreover, previous studies have shown that lung inflammation, caused by cigarette smoke, is associated with increased serum levels of SP-D, probably through increased translocation of this protein into the systemic circulation [25].

Another interesting aspect of our findings is that circulating SP-B correlated significantly and positively with BMI. A relationship between SP-A plasma levels and obesity was detected, suggesting that obesity, with all its possible implications such as chronic hypoxia, could be considered the draining force [26].

### 4.3. Limits

Our study presents some limits, such as the small sample size and the lack of comparison between mature and immature SP-B forms. Given the conflicting data available, the need remains to better clarify the possible role of SP-B and its precursor. Although SP-B is clearly the most specific protein synthesized by alveolar cells, the immature form is the one among the SPs unlikely to be present in the bloodstream under normal circumstances. Indeed, the maturation process of SP-B is a complex multi-step process that happens inside the alveolar epithelial cell, with the highest concentration of the mature form located near the cell air surface, while the immature forms are located inside cell organelles (endoplasmic reticulum and Golgi and multi-vesicular bodies) and released in response to cell membrane damage. Furthermore, smoking status, age, and treatment may be considered as influencing factors.

## 5. Conclusions

SP-B plasma levels are increased in COPD patients and may be considered as a prognostic marker of lung functional decline. The merit of the present study is to analyze the relationship between SP-B and the clinical, functional and radiological aspects in a cohort of COPD patients, underlining its potential role in a clinical scenario. Furthermore, SP-B, given its pulmonary specificity, could also be an early marker of disease.

## Figures and Tables

**Figure 1 biomedicines-11-00124-f001:**
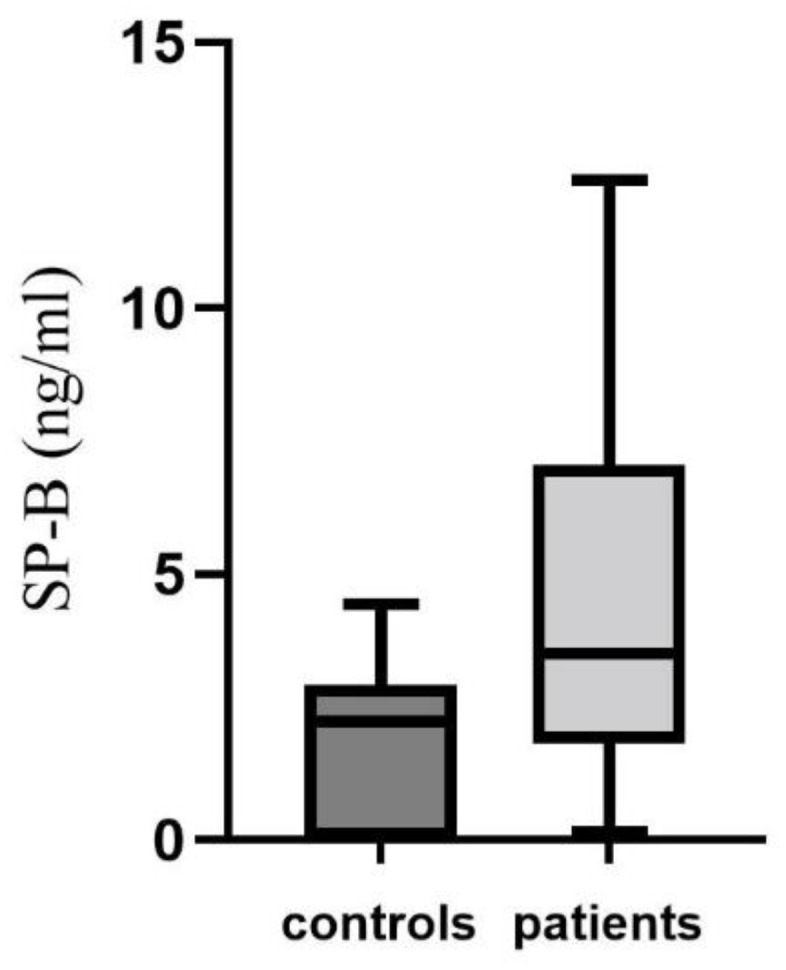
Differences in SP-B plasma levels between healthy smokers and COPD patients.

**Figure 2 biomedicines-11-00124-f002:**
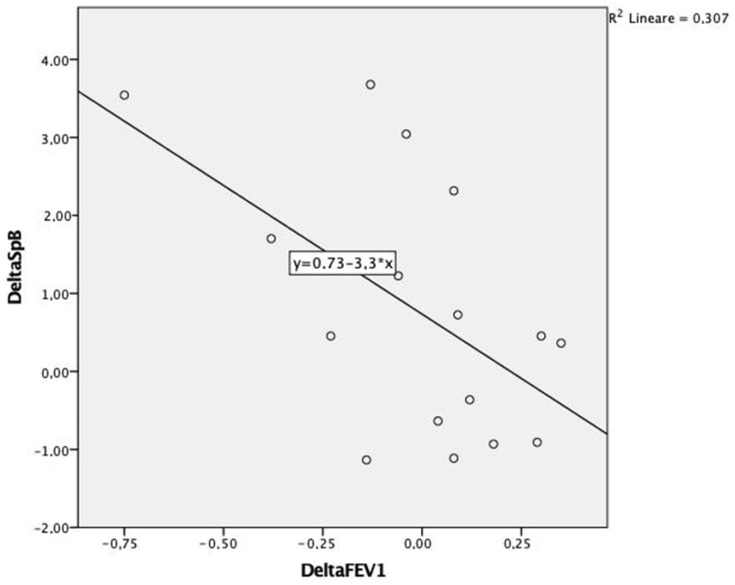
Correlation between delta FEV1 and delta SP-B over the 12-month period (Pearson −0.5; *p* < 0.02).

**Table 1 biomedicines-11-00124-t001:** Demographic and clinical characteristics of the study population.

	COPD *n* = 40	Healthy Smokers *n* = 20	*p* Value
Male/female *n* (%)	30/10 (75/25)	13/7 (65/35)	NS
Age years (mean ± SD)	71 ± 8	65 ± 7	<0.05
BMI (Kg/m^2^)	26.8 ± 5.5	28.2 ± 6.1	NS
CAT (mean ± SD)	13 ± 5	/	
mMRC *n* (%) 0–1 >2	23 (46%) 27 (54%)	/	
Current smoker *n* (%)	14 (35%)	16 (80%)	<0.05
Former smoker *n* (%)	26 (65%)	4 (20%)	<0.05
SP-B plasma levels (ng/mL)	1.78 ± 1.5	4.72 ± 3.2	<0.05

Data are presented as mean ± SE or *n* (%), unless otherwise stated. BMI: body mass index; CAT: COPD Assessment Test; mMRC: Modified British Medical Research Council Questionnaire; SP-B: surfactant protein B.

## Data Availability

The data that support the plots within this paper and other finding of this study are available from the corresponding author upon reasonable request.

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
