# Peer review of "Surfactant Protein B Plasma Levels: Reliability as a Biomarker in COPD Patients"

_biomedicines, 2023, doi:10.3390/biomedicines11010124_

Round 1

Reviewer 1 Report

1) Abstract. Results: A statistically significant difference was found in plasma Sp-B levels between the healthy smokers compared to COPD patients (p<0.001). The change in FEV1 after 12 months (T1-T0) showed a significantly negative correlation respect to the change in Sp-B levels (p<0.05)- Please, add the most important values of parameters  to support the results.

2) Abstract. Conclusions: Surfactant protein B is specific in identifying COPD patients and providing prognostic data on its functional decline. Please, improve the conclusions

3) 1. Introduction L27-32 Chronic Obstructive Pulmonary Disease (COPD) is a common worldwide disease  and is the third leading cause of death [1]. The diagnosis of COPD is based on both clinical  signs such as cough, sputum production, dyspnea, previous exposure to inhalation risk  factors and/or frequent respiratory exacerbations, and functional tests, characterized by  irreversible airflow limitation FEV1/FVC<0,70 post-bronchodilator during forced expira tion test [2]. In order to discuss the previously described points, important references are needed to be added, such as:

A-Definition and Nomenclature of Chronic Obstructive Pulmonary Disease: Time for Its Revision. Am J Respir Crit Care Med. 2022;206(11):1317-1325. doi:10.1164/rccm.202204-0671PP

B-COPD Definition: Is It Time to Incorporate Also the Concept of Lung Regeneration's Failure? [published online ahead of print, 2022 Sep 29]. Am J Respir Crit Care Med. 2022;10.1164/rccm.202208-1508LE. doi:10.1164/rccm.202208-1508LEù

4) Introduction. L51-55. Pulmonary surfactant has important functions beyond reducing surface tension in  order to avoid lung collapse during respiration and provides a crucial first line of defense  against infection [4]. Pulmonary surfactant is a complex mixture of lipids (90%) and pro-  teins (5–10%) that constitutes the mobile liquid phase covering the large surface area of  the alveolar epithelium. Four surfactant proteins (SPs), SP-A, SP-B, SP-C, and SP-D, are  intimately associated with surfactant lipids in the lung. Please add some recent references:

a-Severe COVID-19 ARDS Treated by Bronchoalveolar Lavage with Diluted Exogenous Pulmonary Surfactant as Salvage Therapy: In Pursuit of the Holy Grail?. J Clin Med. 2022;11(13):3577. Published 2022 Jun 21. doi:10.3390/jcm11133577

b-Alveolar type II cells and pulmonary surfactant in COVID-19 era. Physiol Res. 2021;70(S2):S195-S208. doi:10.33549/physiolres.934763

5) L 85-88. The aim of the present exploratory study was to evaluate serum levels of Sp-B in  COPD patients compared to healthy smokers, to explore the relationship between Sp-B  plasma levels and clinical, radiological functional aspects of this pathology and its role as  a prognostic factor. Please, improve the descriptio of study aim.

6) 3. Results L146 3.1. subsection. Please correct the title of the section.

7) 6) 3. Results L146 3.1. subsection.  Please, underline in the results the most important statistical values to support the data.

8) 5. Conclusions L267-270. In conclusion, surfactant protein B is specific in identifying COPD patients and  providing prognostic data on its functional decline, however at present there are many  points to clarify about its possible role as a biomarker. We hope that further studies are  needed to be able to establish if it is really useful as biomarker. Please, improve the description of conclusions, underline the novelty of the study and the possible clinical implications

Reviewer 2 Report

The idea for the work is interesting, but its execution is mediocre.

Introduction. That is quite correct. The aim of the study could be accepted.

Material and methods - I have reservations here. Some wording sounds weird, like "Patients enrolled were 40 selected participants suffering from stable COPD defined by GOLD recommendation and 20 healthy smokers." Too many generalizations.

There are many  grammatical errors, such as (less than 4 week instead of 4 weeks) "We also analyze (it should be analyzed) the correlation between di (? what means "di"?) levels of emphysema on ct score and SPB but no significant correlation was found." "inflammatory protei[20]n-2" - what does it mean?

Results: The record needs improvement. What kind of record is this; is it an average +/- SD? e.g., "Compared the other one (71 + 8 vs 65 + 7) and there was a prevalence of 150 of men (75% vs 65%)". No "vs" but vs.

Table 1. Units are missing in many places, and for example, in age, it is not detailed that it is given in years... all this makes me feel that the article was written very quickly. Are there differences between COPD and Controls in terms of, e.g., age, gender, and sp-B plasma levels? This is unclear from the table; there is no p-value for comparisons. In what units is Sp-B - it is not in the table.

All these issues are not entirely correctly presented.

Table 1. It would be helpful to explain the abbreviations in Abbreviation under the table, e.g., CAT?

In the discussion on page 6, there is number 19 on line 217, and line 219 is number 20. What are they about?; "the toxic effects of cigarette smoke [20]-"

Conclusions. They are very general and contribute little.

To sum up. This article is tough to read. It is written far below accepted standards.
Although the topic is interesting, the presentation of the results obtained and their discussion are poor.

Maybe, the authors should add schemes in the description of the groups. Besides, I recommend dividing the discussion into sections.  

There is a general need to organize the article better.
I recommend that the authors re-read the paper carefully and correct errors, many of which make it impossible to follow the authors' ideas.

Round 2

Reviewer 2 Report

Dear Authors,

your article has been improved and in my opinion can be accepted as it stands (even you haven't highlited the changes you have made).

I wish you a Marry Christmas and a Happy New Year :)

Best regards

D.F.